# Hypovascular tumors developed into hepatocellular carcinoma at a high rate despite the elimination of hepatitis C virus by direct-acting antivirals

**Kazuaki Tabu[1], Seiichi Mawatari[1]\*, Kohei Oda[1‡], Kotaro Kumagai[1‡], Yukiko Inada[2‡], Hirofumi Uto[2‡], Akiko Saisyoji[3‡], Yasunari Hiramine[3‡], Masafumi Hashiguchi[4‡], Tsutomu Tamai[4‡], Takeshi Hori[4‡], Kunio Fujisaki[5‡], Dai Imanaka[6‡], Takeshi Kure[7‡], Ohki Taniyama[1‡], Ai Toyodome[1‡], Sho Ijuin[1‡], Haruka Sakae[1‡], Kazuhiro Sakurai[8‡], Akihiro Moriuchi[8‡], Shuji Kanmura[1‡], Akio Ido[1]**

1 Digestive and Lifestyle Diseases, Department of Human and Environmental Sciences, Kagoshima University Graduate School of Medical and Dental Sciences, Kagoshima, Kagoshima, Japan, 2 Center for Digestive and Liver Diseases, Miyazaki Medical Center Hospital, Miyazaki, Miyazaki, Japan, 3 Department of Internal Medicine, Kagoshima Kouseiren Hospital, Kagoshima, Kagoshima, Japan, 4 Department of Gastroenterology and Hepatology, Kagoshima City Hospital, Kagoshima, Kagoshima, Japan, 5 Department of Hepatology, Kirishima Medical Center, Kirishima, Kagoshima, Japan, 6 Department of Gastroenterology, Ikeda Hospital, Kanoya, Kagoshima, Japan, 7 Department of Gastroenterology, Kagoshima City Medical Association Hospital, Kagoshima, Kagoshima, Japan, 8 Department of Gastroenterology, National Hospital Organization Kagoshima Medical Center, Kagoshima, Kagoshima, Japan

☯ These authors contributed equally to this work.
‡ These authors also contributed equally to this work.
\* mawatari@m2.kufm.kagoshima-u.ac.jp

## Abstract

### Background and aims

Direct-acting antivirals (DAAs) against hepatitis C virus (HCV) exert high anti-HCV activity and are expected to show anti-inflammatory effects associated with HCV elimination. Furthermore, hepatocellular carcinoma (HCC) is known to dedifferentiate from hypovascular tumors, such as dysplastic nodules or well-differentiated HCC, to hypervascular tumors. We therefore explored whether or not DAAs can suppress the growth and hypervascularization of hypovascular tumors.

### Methods

We enrolled 481 patients with HCV genotype 1 infection who were treated with Daclatasvir and Asunaprevir therapy. Of these, 29 patients had 33 hypovascular tumors, which were confirmed by contrast-enhanced MRI or CT before therapy. We prospectively analyzed the cumulative incidence of HCC, i.e. the growth or hypervascularization of hypovascular tumors, and compared the HCC development rates between patients with hypovascular tumors and those without any tumors.

### Results

The mean size of the hypovascular tumors was 11.3 mm. Twenty seven of 29 patients who achieved an SVR had 31 nodules, 19 of 31 nodules (61.3%) showed tumor growth or

**Data Availability Statement:** All relevant data are within the paper and its Supporting Information files.

**Funding:** This work was supported in part by a grant-in-aid from the Ministry of Health, Labour and Welfare of Japan (grant number: 18K15821) to SM and Bristol-Myers Squibb Co., Ltd to AI. The funders had no role in study design, data collection and analysis, decision to publish, or preparation of the manuscript.

**Competing interests:** This work was supported in part by a grant-in-aid from the Ministry of Health, Labour and Welfare of Japan and Bristol-Myers Squibb Co., Ltd. This work used marketed products (Daklinza and Sunvepra) manufactured by Bristol-Myers Squibb Co., Ltd. However, the funders did not have any additional role in the study design, data collection and analysis, decision to publish, or preparation of the manuscript. This does not alter our adherence to PLOS ONE policies on sharing data and materials. AI received honoraria for lectures from Bristol-Myers Squibb Co., Ltd., MSD Co., Ltd., Gilead Sciences Co., Ltd., and Abbvie Inc., and received research funding from Eisai Co., Ltd., Bristol-Myers Squibb Co., Ltd, MSD Co., Ltd., and Abbvie Inc. YH received honoraria from Otsuka Pharmaceutical Co., Ltd. and Asuka Pharmaceutical Co., Ltd.. The other authors declare no conflicts of interest in association with the present study.

hypervascularization, and 12 (38.7%) nodules showed no change or improvement. The cumulative incidence rates of tumor growth or hypervascularization were 19.4% at 1 year, 36.0% at 2 years, 56.6% at 3 years, and 65.3% at 4 years. Among the patients who achieved a sustained virologic response, the cumulative HCC development rates of patients with hypovascular tumors was significantly higher than in those without any tumors. A Cox proportional hazard analysis showed that a history of HCC therapy, the presence of a hypovascular tumor, and AFP >4.6 ng/mL at the end of treatment were independent risk factors for HCC development.

## Conclusion

Hypovascular tumors developed into HCC at a high rate despite the elimination of HCV by DAAs. As patients with hypovascular tumors were shown to have a high risk of HCC development, they should undergo strict HCC surveillance.

## Introduction

Liver cancer is the sixth-most commonly diagnosed cancer and the fourth leading cause of cancer death worldwide [1, 2]. The majority of primary liver cancer is hepatocellular carcinoma (HCC). HCC occurs in patients with underlying liver disease, mostly as a result of hepatitis B or C virus (HBV or HCV) infection or alcohol abuse [2]. The persistent inflammation caused by HCV infection causes liver fibrosis, leading to cirrhosis and HCC [3]. One of the goals of therapy is to cure HCV infection in order to prevent the complications of HCV-related liver diseases, including hepatic necroinflammation, fibrosis, cirrhosis, decompensated cirrhosis, HCC, and death [4].

HCC occurs as a result of hepatic inflammation and/or changes in the tumor microenvironment [2]. Interferon (IFN)-based therapy for chronic hepatitis C provides therapeutic benefits, such as the suppression of progressive fibrosis via HCV elimination, thereby preventing HCC [5, 6]. As a mechanism for suppressing hepatocarcinogenesis by IFN, the involvement of a direct antiviral effect and tumor-suppressive action through the induction of antitumor immunity have been considered [7, 8].

Direct-acting antivirals (DAAs) against HCV exert high anti-HCV activity via direct action on the viral protein to inhibit viral growth and replication, and are expected to show anti-inflammatory effects associated with HCV elimination [1]. Recently, some reports shows that DAA also prevent hepatocarcinogenesis in a manner similar to IFN [9–12].

Furthermore, HCC is known to dedifferentiate from hypovascular liver tumors (dysplastic nodules and well-differentiated HCC) to hypervascular tumors [13, 14]. With this rationale, we therefore examined whether or not DAAs can suppress the growth and hypervascularization of hypovascular liver tumors.

## Materials and methods

### Patients

Fig 1 shows the flow chart of study patient enrollment. We enrolled 481 patients with HCV genotype 1 infection who were treated with daclatasvir (DCV; Daklinza; Bristol-Myers Squibb, New York, NY, USA) at 60 mg/day once daily and asunaprevir (ASV; Sunvepra; Bristol-Myers Squibb) at 200 mg/day twice daily for 24 weeks, including patients who had previously

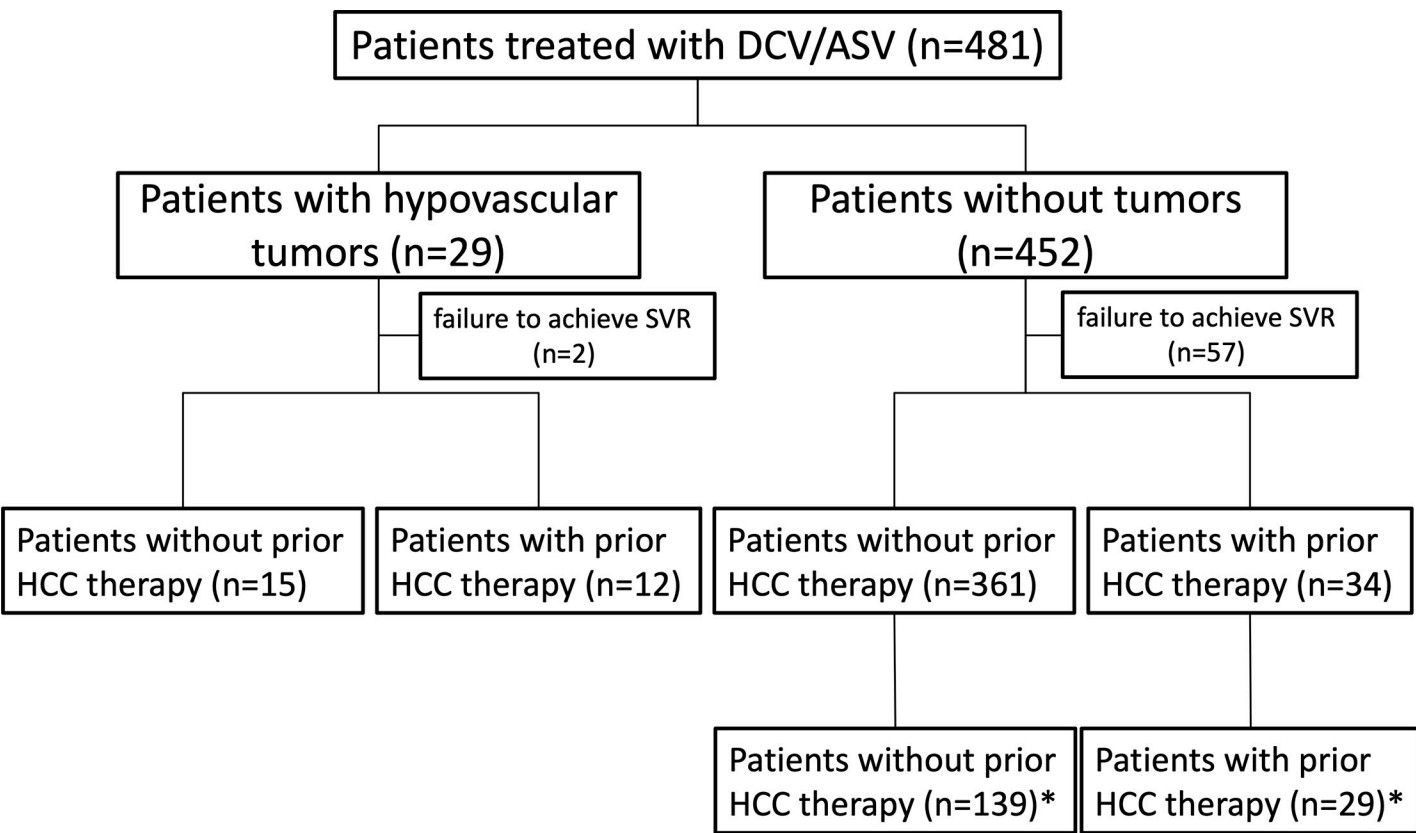

**Fig 1. Flow chart of study patient enrollment.** *underwent contrast-enhanced MRI or CT within two years before therapy or during DAA therapy.

undergone curative HCC therapy such as surgical resection or radio frequency ablation (RFA) and discontinued therapy due to adverse events, at 21 facilities belonging to the Kagoshima Liver Study Group in Japan, between September, 2014, and December, 2016. A sustained virologic response (SVR) was defined as undetectable HCV RNA at 24 weeks after the end of treatment.

Among these patients, 29 patients had hypovascular tumors that were confirmed by contrast-enhanced (CE) MRI or CT before therapy. These surveys were conducted approximately 50 days before therapy. They were prospectively observed every three to four months to assess the tumor size and hypervascularization status. HCC development was defined by 20% growth or hypervascularization of a hypovascular nodule. We analyzed the cumulative incidence rates of growth or hypervascularization of hypovascular nodules. In contrast, 452 patients were confirmed to be tumor-free by ultrasound sonography (US), CT, or MRI before DAA therapy and underwent HCC surveillance according to Japanese guidelines. These guidelines state that cirrhotic patients have an extremely high risk of developing HCC and should be monitored every three to four months and that non-cirrhotic patients have a high risk of developing HCC and should be monitored every six months by US, CT, or MRI [15]. HCC was diagnosed when typical vascular findings were observed by CE-CT or MRI, i.e. hyper-enhancement in the arterial phase and a washout pattern in the portal or delayed phase.

We analyzed the respective cumulative HCC development rates of patients with hypovascular tumors and those without any tumors. Among the patients who achieved an SVR, 168 without a hypovascular tumor underwent CE-MRI or CT within 2 year before therapy or during

DAA therapy. We compared the cumulative HCC development rates between patients with and without hypovascular tumors. The Fib-4 index, a surrogate marker of liver fibrosis, was calculated based on the methods of a previous study [16]. The initiation of the observation period was defined as the initiation of DAA treatment.

The study protocol conformed to the ethical guidelines of the Declaration of Helsinki and was approved by Kagoshima University Hospital and the research ethics committee of each participating facility on December 10, 2014, and October 21, 2015 (approval number: 26–142, 26–143, and 27–118). Written informed consent was obtained from each patient.

## Definition of hypovascular tumor

Hypovascular tumors were diagnosed by gadolinium ethoxybenzyl diethlenetriamine pentaacetic acid (Gd-EOB-DTPA) CE-MRI based on scans showing no intense enhancement in the arterial phase but a low signal intensity indicating a hypovascular liver tumor (i.e. dysplastic nodules and well-differentiated HCC) in the hepatobiliary phase or CE-CT showing no enhancement in the arterial phase but a low density in the arterial, portal, or delayed phase.

## Detection of HCV resistance-associated substitutions (RAS)

As described in detail previously, we investigated the viral genome sequence by direct sequencing [17–19]. The nucleotide sequences of the second amplicons were determined using a Big-Dye Terminator v3.1 Cycle Sequencing Kit (Thermo Fisher Scientific, Waltham, MA, USA) and Sanger sequencing. The sequences of the non-structural (NS) 3 and NS5A RAS positions in the HCV gene were determined using HCV-Con1 (accession no. AJ238799)

## Statistical analyses

Statistical analyses were performed using the IBM Statistical Package for Social Sciences (SPSS) software program (version 22 IBM SPSS Statistics, Armonk, NY, USA). Categorical data were compared using the chi-squared test and Fisher's exact test, as appropriate. Continuous variables were analyzed using the Mann-Whitney U test. The Kaplan–Meier method and log rank test were used to analyze the cumulative HCC development rates. P values of $<0.05$ were considered to indicate statistical significance. Factors associated with HCC development were determined using a Cox proportional hazards analysis with forward selection using $p<0.10$ as a cutoff for inclusion in the model. For the categorical data, we determined the cut-off values at which the optimal sensitivity and specificity were achieved using receiver operating characteristic curves.

## Results

### Baseline characteristic of patients with hypovascular tumor

The baseline characteristic of patients with hypovascular tumors showed that the mean age was 73 years old, 34.5% were male, 79.3% had liver cirrhosis, and 41.4% had a history of HCC therapy. Thirty-three tumors were detected in 29 patients, and the mean hypovascular tumor size was 11.3 mm (Table 1). No patients had RAS of NS5A L31 or Y93 or NS3 D168 at baseline. Twenty-seven patients (93.1%) achieved an SVR.

On comparing the baseline characteristics between patients with hypovascular tumors and those without any tumors, the patients with hypovascular tumors tended to be older; more frequently had liver cirrhosis and a history of HCC therapy; and had higher values of total bilirubin, hyaluronic acid, Fib-4 index, and AFP before therapy and at the end of treatment and lower platelet counts and albumin levels than those without any tumors (Table 1).

**Table 1. Baseline characteristic of patients with hypovascular tumors and without any tumor.**

|  | Total (n = 481) | tumor(-) (n = 452) | Hypovascular tumor(+) (n = 29) | *P* value |
|---|---|---|---|---|
| Age, years | 68.8±9.1 | 68.5±9.2 | 73.1±5.5 | 0.007 |
| Male, n (%) | 193 (40.1) | 183 (40.5) | 10 (34.5) | 0.332 |
| Liver cirrhosis, n (%) | 133 (27.7) | 110 (24.3) | 23 (79.3) | <0.001 |
| History of interferon-based therapy, n (%) | 255 (53.0) | 236 (52.2) | 19 (65.5) | 0.373 |
| History of HCC therapy, n (%) | 53 (11.0) | 41 (9.1) | 12 (41.4) | <0.001 |
| Platelets, ×$10^4$/μL (n = 480) | 13.6±5.5 | 13.9±5.4 | 9.6±5.0 | <0.001 |
| Total bilirubin, mg/dL (n = 480) | 0.8±0.4 | 0.8±0.4 | 1.0±0.5 | 0.029 |
| ALT, U/L | 49±37 | 49±38 | 53±32 | 0.088 |
| GGT, U/L (n = 480) | 44±42 | 43±43 | 47±28 | 0.130 |
| Albumin, g/dL (n = 468) | 4.0±0.4 | 4.0±0.4 | 3.8±0.6 | 0.007 |
| Hyaluronic acid, ng/mL (n = 413) | 225±389 | 214±383 | 431±446 | 0.001 |
| Fib-4 index (n = 480) | 4.69±3.33 | 4.50±3.21 | 7.52±3.93 | <0.001 |
| AFP (Before), ng/mL (n = 479) | 12.4±30.3 | 11.8±30.4 | 22.6±26.1 | <0.001 |
| AFP (End of treatment), ng/mL (n = 453) | 6.0±12.8 | 5.8±12.7 | 9.0±12.7 | 0.017 |
| DCP, mAU/mL (n = 375) | 21.4±11.2 | 20.8±10.1 | 28.7±20.5 | 0.069 |
| SVR, n (%) | 422 (87.7) | 395 (87.4) | 27 (93.1) | 0.284 |
| Hypovascular tumor size, mm (n = 33) | - | - | 11.3±3.7 | - |
| HCC development, n (%) | 65 (13.5) | 44 (9.7) | 21 (72.4) | <0.001 |

Data are shown as the mean ± standard deviation. HCC, hepatocellular carcinoma; ALT, alanine transaminase; GGT, γ-glutamyltransferase; AFP, α-fetoprotein; DCP, des-γ-carboxy prothrombin; SVR, sustained virologic response

## Outcomes of hypovascular tumors in patients who achieved an SVR

Fig 2 shows the outcomes of hypovascular tumors in patients who achieved an SVR. In 27 of 29 patients with hypovascular tumors who achieved an SVR, 4 patients had 2 hypovascular tumors each, resulting in 31 hypovascular tumors in total. Of these 31 nodules, 19 (61.3%) showed tumor growth or hypervascularization, while 12 (38.7%) showed no change or improvement. Among the patients with two hypovascular tumors, both tumors in two patients showed no change, while the tumors in the other two patients showed hypervascularization in one nodule and no change in the other. Two patients developed HCC in other lesions, but the hypovascular tumor did not change. Three nodules were treated by surgery, nine were treated by RFA, and seven were treated by transcatheter arterial chemoembolization (TACE). No statistically significant differences were observed between tumor growth or hypervascularization and no change or improvement regarding the hypovascular tumor size (10.4±2.5 mm vs. 12.2±4.7 mm, p = 0.318).

## The cumulative incidence rates of growth or hypervascularization of hypovascular tumors

In patients who achieved an SVR, the cumulative incidence rates of growth or hypervascularization of hypovascular tumors were 19.4% at 1 year, 36.0% at 2 years, 56.6% at 3 years, and 65.3% at 4 years (Fig 3A). In addition, the respective cumulative HCC development rates of patients with and without a history of curative HCC therapy were 23.1% and 16.7% at 1 year, 55.1% and 22.2% at 2 years, 73.1% and 44.5% at 3 years, and 82.1% and 52.5% at 4 years, respectively (Fig 3B). There was no significant difference in patients with and without a history of curative HCC therapy (p = 0.113) (Fig 3B). In addition, there were no patients who had unexpected tumor growth during the observation periods.

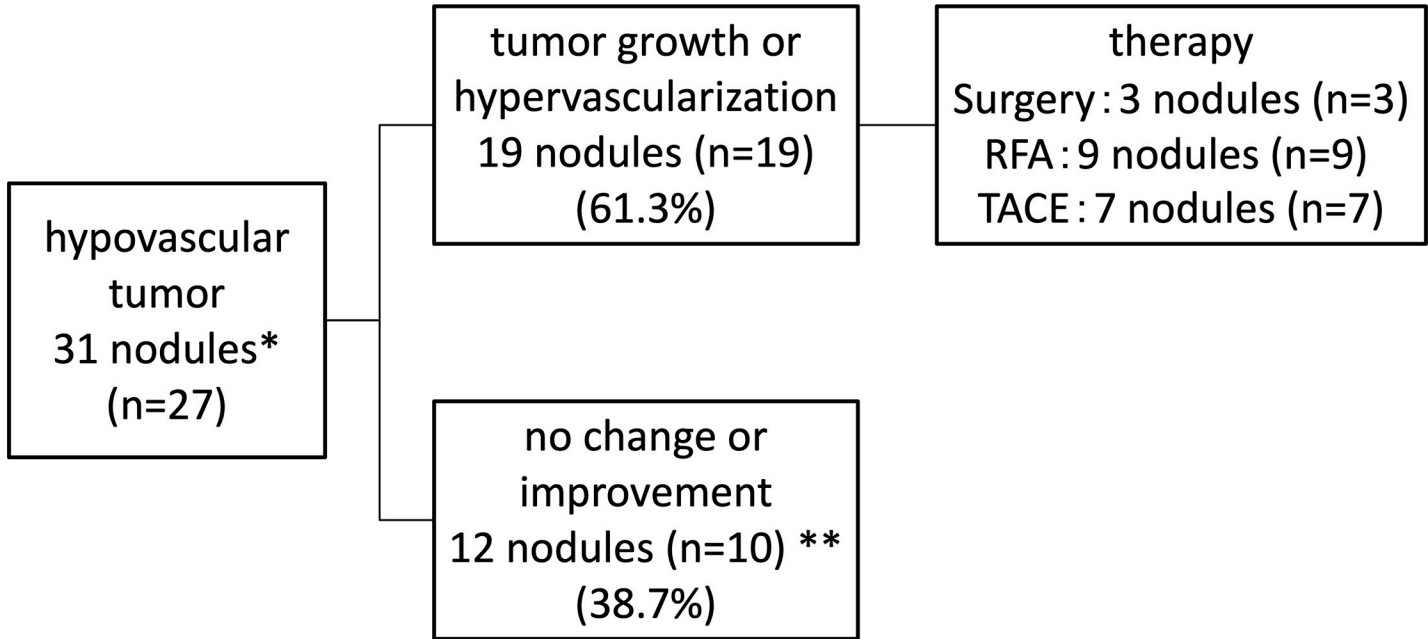

**Fig 2. Outcome of hypovascular tumors in patients who achieved an SVR.** *Four patients had two hypovascular tumors each, and in two of these patients, one nodule showed hypervascularization, while the other nodule showed no change. **Two patients developed HCC in other lesions, but the hypovascular tumor did not change.

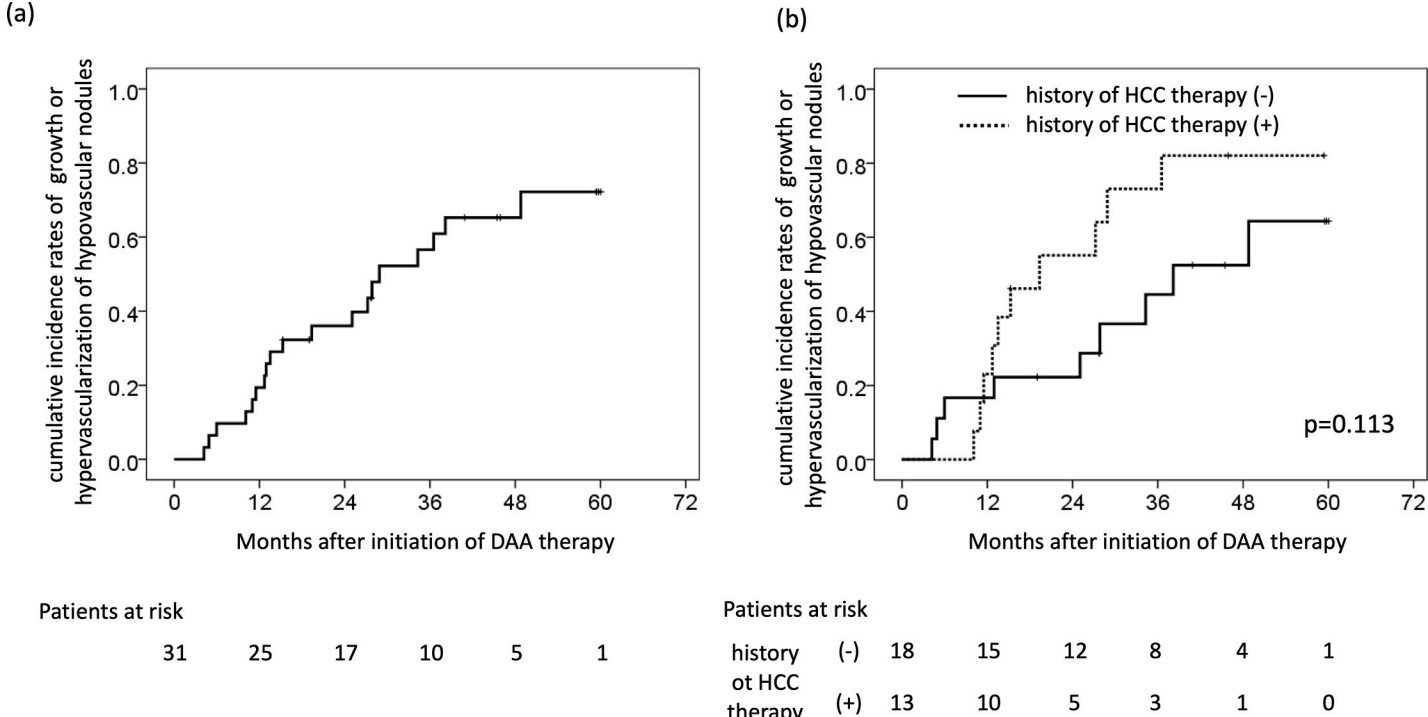

**Fig 3. Cumulative incidence rates of growth or hypervascularization of hypovascular tumor.** (a) Cumulative incidence rates of growth or hypervascularization of hypovascular tumors. (b) Cumulative incidence rates in patients with or without a history of HCC therapy.

## The comparison of the incidence of HCC between patients with hypovascular tumors and without any tumors who achieved an SVR

We compared the incidence of HCC between patients with hypovascular tumors and those without any tumors who achieved an SVR. Overall, 422 patients (87.7%) achieved an SVR (including 36 patients discontinued therapy), and 56 developed HCC (Table 2). Among patients who achieved an SVR, a univariate analysis showed that patients who developed HCC were more frequently male and had liver cirrhosis, a history of HCC therapy, and lower platelet and albumin values and higher ALT, GGT, hyaluronic acid, fib-4 index, and AFP values before and at the end of treatment (Table 2).

We performed a multivariate analysis using factors that were significantly different in the univariate analysis as covariates. A Cox proportional hazard analysis showed that a history of HCC therapy, the presence of hypovascular tumor, and an AFP >4.6 ng/mL at the end of treatment were independent risk factors for HCC development (Table 3). Among patients without any tumors who achieved an SVR, 168 underwent CE-MRI or CT within 2 years before therapy or during DCV+ASV therapy (Fig 1). The patients who underwent CT or MRI before DAA therapy had higher rates of HCC development than those who underwent US alone. Because the patients who underwent CT or MRI more frequently had liver cirrhosis and a history of HCC therapy, had higher total bilirubin, GGT, hyaluronic acid, Fib-4 index, and AFP values before therapy and at the end of treatment, and had lower platelet counts and albumin levels than those who underwent US alone (S1 Table). We performed a Cox proportional hazard analysis in patients who underwent CE-MRI or CT including hypovascular tumors, and similar results were obtained (Table 3).

## The comparison of the incidence of HCC between patients with or without hypovascular tumors and a history of HCC therapy who achieved an SVR

We compared the cumulative HCC development rates of patients with or without a hypovascular tumor and the history of HCC therapy. The patients were classified into four groups

**Table 2. Baseline characteristics of patients who achieved an SVR.**

|  | Total (n = 422) | HCC development(-) (n = 366) | HCC development(+) (n = 56) | P value |
|---|---|---|---|---|
| Age, years | 68.8±9.2 | 68.5±9.5 | 70.8±7.4 | 0.120 |
| Male, n (%) | 177 (41.9) | 147 (40.2) | 30 (53.6) | 0.041 |
| Liver Cirrhosis, n (%) | 109 (27.7) | 79 (21.6) | 30 (53.6) | <0.001 |
| History of interferon-based therapy, n (%) (n = 421) | 218 (51.7) | 181 (49.5) | 37 (66.1) | 0.066 |
| History of HCC therapy, n (%) | 46 (10.9) | 19 (5.2) | 27 (48.2) | <0.001 |
| Presence of hypovascular tumor, n (%) | 27 (6.4) | 7 (1.9) | 20 (35.7) | <0.001 |
| Platelets, ×$10^4$/μL | 13.7±5.5 | 14.1±5.4 | 11.2±5.4 | <0.001 |
| Total bilirubin, mg/dL | 0.8±0.4 | 0.8±0.3 | 0.9±0.4 | 0.223 |
| ALT, U/L | 49±38 | 49±39 | 53±28 | 0.009 |
| GGT, U/L (n = 421) | 42±36 | 40±32 | 59±55 | 0.002 |
| Albumin, g/dL (n = 412) | 4.0±0.4 | 4.0±0.4 | 3.8±0.5 | <0.001 |
| Hyaluronic acid, ng/mL (n = 359) | 230±411 | 213±390 | 347±526 | <0.001 |
| Fib-4 index | 4.67±3.40 | 4.40±3.27 | 6.44±3.72 | <0.001 |
| AFP (Before), ng/mL (n = 420) | 11.4±22.9 | 9.5±16.1 | 23.6±45.5 | <0.001 |
| AFP (End of treatment), ng/mL (n = 399) | 5.7±12.3 | 5.1±12.1 | 9.2±12.9 | <0.001 |
| DCP, mAU/mL (n = 327) | 21.7±11.7 | 20.9±11.0 | 26.2±14.5 | 0.069 |

Data are shown as the mean ± standard deviation.

HCC, hepatocellular carcinoma; ALT, alanine transaminase; GGT, γ-glutamyltransferase; AFP, α-fetoprotein; DCP, des-γ-carboxy prothrombin; SVR, sustained virologic response

**Table 3. Factors associated with HCC development.**

| Category | Cut off | Multivariate analysis | | |
|---|---|---|---|---|
| | | Hazard Ratio | 95% CI | *P* value |
| *Among the patients who achieved SVR* (n = 331) | | | | |
| Sex | Male | 1.754 | 0.959–3.210 | 0.068 |
| History of HCC therapy | Yes | 6.947 | 3.685–13.099 | <0.001 |
| Hypovascular tumor | Presence | 5.119 | 2.566–10.210 | <0.001 |
| AFP level at end of treatment | >4.6 ng/mL | 2.377 | 1.298–4.353 | 0.005 |
| *Among the patients who achieved SVR and underwent contrast-enhanced MRI or CT within 2 year before therapy or during DAA therapy* (n = 156) | | | | |
| History of HCC therapy | Yes | 5.200 | 2.671–10.125 | <0.001 |
| Hypovascular tumor | Presence | 4.097 | 2.038–8.237 | <0.001 |
| AFP level at end of treatment | >4.6 ng/mL | 2.133 | 1.105–4.116 | 0.024 |

Other covariates were the age, presence of cirrhosis, platelet, alanine transaminase, γ-glutamyltransferase, and hyaluronic acid, albumin, AFP before therapy, and Fib-4 index.

HCC, hepatocellular carcinoma; 95% CI, 95% confidence interval; HCC, hepatocellular carcinoma; SVR, sustained virologic response; MRI, magnetic resonance imaging; CT, computed tomography; DAA, direct acting antivirals; AFP, alfa-fetoprotein

according to the presence of a hypovascular tumor and the history of HCC therapy, and the cumulative HCC development rates were examined in each group. The four groups were as follows: (a) no hypovascular tumor and no history of HCC therapy, (b) no hypovascular tumor and a history of HCC therapy, (c) a hypovascular tumor and no history of HCC therapy, and (d) a hypovascular tumor and a history of HCC therapy.

The cumulative HCC development rates at 1, 2, 3, and 4 years were as follows: group a = 0.6%, 2.8%, 3.8%, 4.8%; group b = 8.8%, 29.4%, 38.4%, 42.5%; group c = 26.7%, 33.3%, 50.6%, 73.7%; and group d = 41.7%, 66.7%, 83.3%, 91.7%, respectively (Fig 4A). Groups b, c, and d showed significantly higher cumulative HCC development rates than group a, and group d showed a significantly higher cumulative HCC development rate than group b (Fig 4A).

Similarly, we analyzed patients who underwent CE-MRI or CT. The cumulative HCC development rates at 1, 2, 3, and 4 years were as follows: group a = 1.4%, 4.4%, 5.9%, 8.3%; group b = 10.3%, 34.5%, 41.4%, 45.9%; group c = 26.7%, 33.3%, 50.6%, 73.7%; and group d = 41.7%, 66.7%, 83.3%, 91.7%, respectively (Fig 4B). Groups b, c, and d showed significantly higher cumulative HCC development rates than group a, and group d showed a significantly higher cumulative HCC development rate than group b (Fig 4B).

## Discussion

In the present study, we investigated the cumulative incidence rates of growth or hypervascularization of hypovascular tumors in 29 patients who were infected with HCV and treated by DAAs. The present study has novelty in the part that examined the detailed clinical course of hypovascular tumors. In patients who achieved an SVR, the cumulative incidence rates of growth or hypervascularization of hypovascular tumors were 19.4% at 1 year, 36.0% at 2 years, 56.6% at 3 years, and 65.3% at 4 years (Fig 3A). In previous studies, the cumulative incidence rates of hypervascularization were 14.9%-25% at 1 year and 45.8%-51% at 2 years [20–22], especially, high-grade dysplastic nodule had high HCC development rates [14]. Toyoda et al. showed the rates of hypervascularization of hypovascular tumors did not differ markedly between the study patients who achieved an SVR and the propensity score-matched patients persistent HCV infection [23]. Compared with previous reports, the cumulative incidence

(a)

(b)

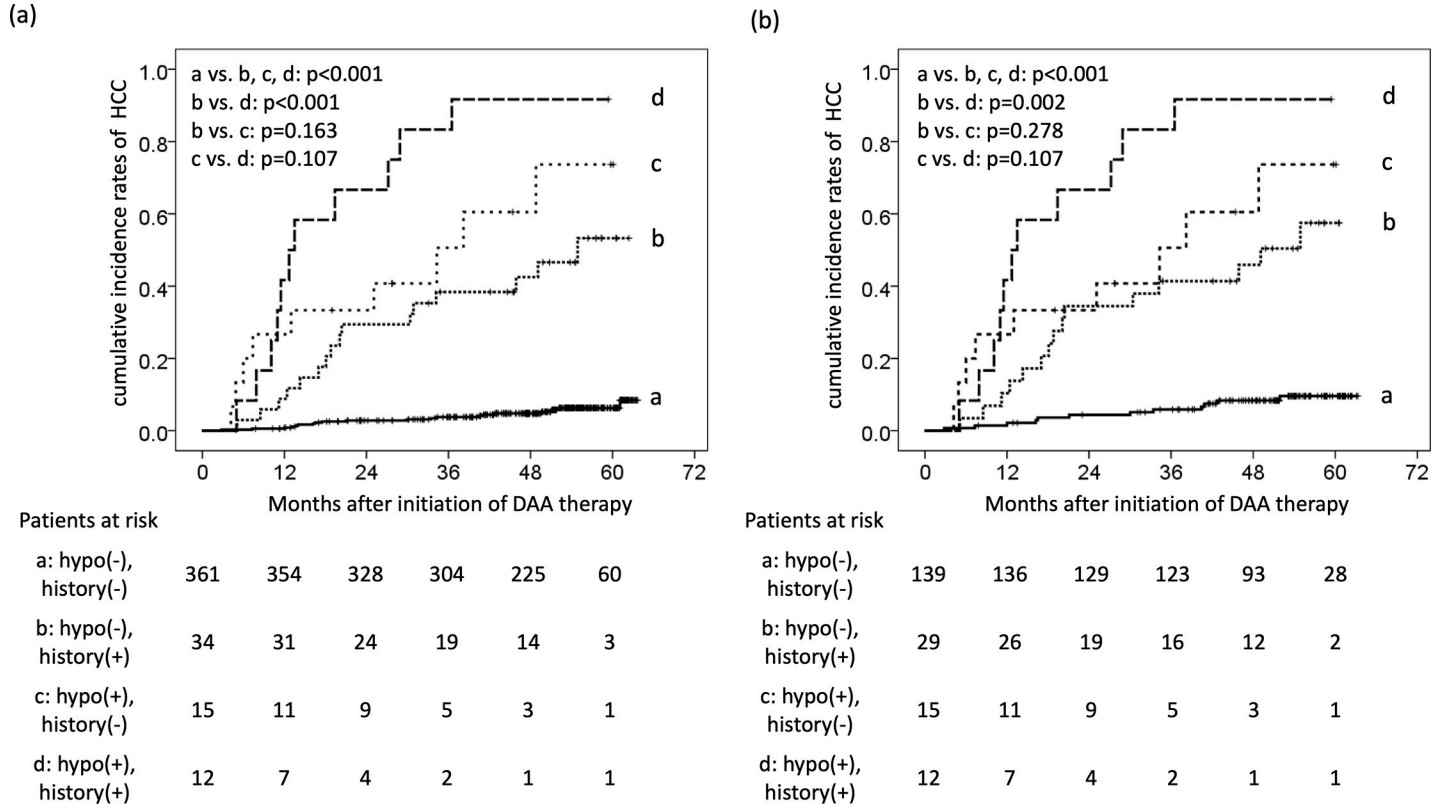

**Fig 4. The comparison of the incidence of HCC between patients with or without hypovascular tumors and a history of HCC therapy who achieved SVR.** (a) Cumulative incidence rates of HCC among the four groups. (b) Cumulative incidence rates of HCC among the four groups in patients who underwent contrast-enhanced MRI or CT within two year before therapy or during DAA therapy. Group a, no hypovascular tumor and no history of HCC therapy; group b, no hypovascular tumor and a history of HCC therapy; group c, a hypovascular tumor and no history of HCC therapy; and group d, a hypovascular tumor and a history of HCC therapy.

rates of hypervascularization were not markedly different in the present study, although the background characteristics were different, including patients with etiologies other than HCV. In other words, DAAs cannot suppress the growth and hypervascularization of hypovascular tumors.

In addition, four patients had two hypovascular tumors each, and in two of these patients, one nodule showed hypervascularization, while the other nodule showed no change (Fig 2). It was considered that these two tumors in these patients had multicentric growth and a different carcinogenic potential in each tumor. In contrast, two patients developed small HCC in other lesions, but their hypovascular tumors showed no change (Fig 2). Of these two patients, one patient with a history of HCC therapy might have developed HCC due to intrahepatic metastasis. However, the other patient did not have any history of HCC. Therefore, the eradication of HCV by DAA therapy may not influence the hypervascularization of hypovascular tumors. However, there are some unclear points concerning the hypervascularization of hypovascular tumors, so the factors associated with the hypervascularization of hypovascular tumors will need to be clarified in a large study population in the future.

Several previous studies have shown the cumulative HCC developing rate in HCV-infected patients with hypovascular tumors who were treated with DAAs. Toyoda et al. showed that, among 164 patients with HCV cirrhosis who achieved an SVR by DAA therapy just before the start of therapy, 38 had non-hypervascular hypointense nodules (NHHNs) on EOB-MRI, and

hypervascularization of NHHNs was observed in 17 patients (44.7%) [24]. Similarly, Ooka et al. showed that, in 864 patients with HCV infection across 2 cohorts, 41 patients developed HCC within 1 year after DAA therapy, and the factor associated with 1-year HCC occurrence and recurrence was the existence of a "dysplastic nodule" on imaging [25]. Marino et al. showed that, in 1,123 patients with cirrhosis who were treated with DAAs, 80 (7%) had non-characterized nodules, and the risk of HCC was significantly increased in patients with non-characterized nodules at baseline [26].

The common findings of the above reports were that patients with hypovascular tumors developed an early incidence of HCC. In the present study, the HCC development rates were higher in patients with hypovascular tumors than in those without any tumors and were similar to the rates in patients without any tumors who had a history of HCC therapy, although the background characteristics were different (Fig 4A and 4B). Furthermore, the patients with hypovascular tumors who had a history of HCC therapy had the highest incidence of HCC. Previous reports were observational studies conducted for one to two years after treatment, but the present study were observed for three to four years after DAA treatment. Since patients with hypovascular tumors have a high hepatocarcinogenetic potential, we should conduct periodic follow-up, such as tumor marker measurements and imaging, for a long time.

Recently, the elimination of HCV induced by DAA was shown to reduce the risk of HCC development and mortality in a large-scale observational study [10, 11, 27, 28]. In addition, the risk of HCC recurrence with DAA treatment is not markedly different from that with IFN treatment [12, 29, 30]. However, it has also been reported that DAA treatment may cause an unexpected onset of HCC [31, 32]. Although the mechanism underlying this unexpected onset of HCC has not been fully clarified, viral elimination by DAA treatment might reduce IFN-alfa-induced intrahepatic immunity, thus resulting in a rapid decrease in natural killer (NK) cell activity and the normalization of the NK cell cytotoxic effector function [33]. In addition, DAAs rapidly reduce inflammation but increase serum vascular endothelial growth factor (VEGF) levels [34], and the DAA-mediated increase in VEGF favors HCC recurrence/occurrence in susceptible patients with more severe fibrosis and splanchnic collateralization who already have abnormal activation in liver tissues of neo-angiogenetic pathways, like angiopoie-tin-2 [35]. Such immunological changes may be associated with a reduction in the immuno-surveillance mechanism of neoplastic clones and an increase in VEGF and may thus promote hepatocarcinogenesis.

The present study was associated with several limitations. First, the number of study patients was small. Second, there were several missing values in the multivariate analysis, so the number of patients who were able to be examined was reduced. Third, various potentially influential factors (alcohol intake, obesity, etc.) were not examined after treatment. Fourth, 29 patients had hypovascular tumors that were confirmed by CE-MRI or CT. These surveys were conducted approximately 50 days before therapy. Further, 452 patients received DAA therapy after they were confirmed to be tumor-free by US, CT, or MRI. However, we did not confirm the exact US survey points before therapy. Thus, the start of the observation period was defined as the time at which DAA treatment was initiated. In addition, 168 patients without a hypovascular tumor underwent CE-MRI or CT within 2 years before therapy or during DAA therapy. Unfortunately, there were few patients who underwent CT or MRI immediately before DAA therapy. We cannot rule out the possibility of growth or hypervascularization of hypovascular tumors, or HCC development in patients who were tumor-free before DAA therapy. Thus, a future study should analyze a larger cohort which imaging surveillance was performed immediately before DAA therapy.

We concluded that hypovascular tumors developed into HCC at a high rate despite the elimination of HCV by DAAs. This suggests that DAAs cannot suppress the growth and

hypervascularization of hypovascular tumors. As patients with hypovascular tumors tend to be older and frequently have liver cirrhosis along with a high risk of HCC development, they should undergo HCC surveillance carefully, similar to patients with a history of HCC therapy.

## Supporting information

**S1 Table. The baseline characteristics of patients and rate of HCC development for each survey method.**
(DOCX)

**S2 Table. Analysis data set.** All patients' data sets were included in the following file.
(XLSX)

**S3 Table. Analysis data set.** All data of hypovascular tumor data sets were included in the following file.
(XLSX)

## Acknowledgments

The present study was carried out in the following 21 facilities (Kagoshima Liver Study Group): Kagoshima University Hospital, Kirishima Medical Center, Miyazaki Medical Center Hospital, Kagoshima Kouseiren Hospital, Kagoshima City Hospital, Saiseikai Sendai Hospital, Kohshinkai Ogura Hospital, Ikeda Hospital, Izumi General Medical Center, Oshima Hospital, Ibusuki Medical Center, Kagoshima medical center, Hirono Clinic, Kagoshima Teishin Hospital, Satsunan Hospital, Nagaki Clinic, Dr. NAKANISHI's office, Southern Region Hospital, Tanegashima Medical Center, Fujimoto General Hospital, and Nakayama Clinic. We thank the following investigators: Shuzo Tashima (Kirishima Medical Center), Yasushi Imamura (Kagoshima Kouseiren Hospital), Toshihiro Fujita (Oshima Hospital), Akihiko Oshige (Ibusuki Medical Center), Shuichi Hirono (Hirono Clinic), Masahito Nagaki (Nagaki Clinic), Chihiro Nakanishi (Dr. NAKANISHI's office), and Toshihiro nakayama (Nakayama Clinic). We also thank Ms. Hiromi Eguchi, Ms. Mayumi Ono, Ms. Etsuko Horiguchi, Ms. Yuko Morinaga, and Ms. Eriko Koreeda for their technical assistance and data management.

The study was presented as the previous draft in The International Liver Congress™ 2019 poster presentations.

## Author Contributions

**Conceptualization:** Kazuaki Tabu, Seiichi Mawatari, Akio Ido.

**Data curation:** Kazuaki Tabu, Seiichi Mawatari, Kohei Oda, Kotaro Kumagai, Yukiko Inada, Hirofumi Uto, Akiko Saisyoji, Yasunari Hiramine, Masafumi Hashiguchi, Tsutomu Tamai, Takeshi Hori, Kunio Fujisaki, Dai Imanaka, Takeshi Kure, Ohki Taniyama, Ai Toyodome, Sho Ijuin, Haruka Sakae, Kazuhiro Sakurai, Akihiro Moriuchi, Shuji Kanmura, Akio Ido.

**Formal analysis:** Kazuaki Tabu, Seiichi Mawatari, Akio Ido.

**Funding acquisition:** Seiichi Mawatari, Akio Ido.

**Investigation:** Kazuaki Tabu, Seiichi Mawatari, Kohei Oda, Kotaro Kumagai, Yukiko Inada, Hirofumi Uto, Akiko Saisyoji, Yasunari Hiramine, Masafumi Hashiguchi, Tsutomu Tamai, Takeshi Hori, Kunio Fujisaki, Dai Imanaka, Takeshi Kure, Ohki Taniyama, Ai Toyodome, Sho Ijuin, Haruka Sakae, Kazuhiro Sakurai, Akihiro Moriuchi, Shuji Kanmura, Akio Ido.

**Methodology:** Kazuaki Tabu, Seiichi Mawatari.

**Project administration:** Kazuaki Tabu, Seiichi Mawatari.

**Supervision:** Akio Ido.

**Validation:** Kazuaki Tabu, Seiichi Mawatari, Akio Ido.

**Visualization:** Kazuaki Tabu, Seiichi Mawatari.

**Writing – original draft:** Kazuaki Tabu, Seiichi Mawatari.

**Writing – review & editing:** Kazuaki Tabu, Seiichi Mawatari, Akio Ido.

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
