## [Decision Letter · Decision Letter 0]

17 Jun 2020

PONE-D-20-16242

Hypovascular tumors developed into hepatocellular carcinoma at a high rate despite the elimination of hepatitis C virus by direct-acting antivirals.

PLOS ONE

Dear Dr. Seiichi Mawatari,

Thank you for submitting your manuscript to PLOS ONE. After careful consideration, we feel that it has merit but does not fully meet PLOS ONE’s publication criteria as it currently stands. Therefore, we invite you to submit a revised version of the manuscript that addresses the points raised during the review process.

We look forward to receiving your revised manuscript.

Kind regards,

Tatsuo Kanda, M.D., Ph.D.

Academic Editor

PLOS ONE

Journal Requirements:

2. During our internal review, we noticed you have overlapping text with a copyrighted abstract of your work published here:

https://www.journal-of-hepatology.eu/article/S0618-8278(19)31480-X/pdf

PLOS ONE cannot (re)publish material without sufficient permission from the original copyright holder to publish under a CC BY license. Please complete one of the following:

(i) Please provide proof that the owner of the content (a) has given you written permission to use it, and (b) has approved of the CC BY license being applied to their content. You may have the following form completed by the owner as proof: https://journals.plos.org/plosone/s/file?id=7c09/content-permission-form.pdf. Alternatively, you may electronically request permissions through from the copyright holder and send us proof of approval, as long as the approval clearly shows that the owner has approved of the CC BY license being applied to their content. Please see https://journals.plos.org/plosone/s/licenses-and-copyright for more information.

OR

(ii) Please cite the abstract and rephrase any duplicated text.

3. Our internal editors have looked over your manuscript and determined that it is within the scope of our Liver Diseases Call for Papers. This collection of papers is headed by a team of Guest Editors for PLOS ONE. Additional information can be found on our announcement page: https://collections.plos.org/s/liver-diseases

If you would like your manuscript to be considered for this collection, please let us know in your cover letter and we will ensure that your paper is treated as if you were responding to this call.

If you would prefer to remove your manuscript from collection consideration, please specify this in the cover letter.

4. Thank you for stating the following in the Financial Disclosure  section:

'This work was supported in part by a grant-in-aid from the Ministry of Health, Labour and Welfare of Japan (grant number: 18K15821) and Bristol-Myers Squibb Co., Ltd. The funders had no role in study design, data collection and analysis, decision to publish, or preparation of the manuscript.'

We note that you received funding from a commercial source: Bristol-Myers Squibb Co., Ltd

Reviewers' comments:

Reviewer's Responses to Questions

**Comments to the Author**

1. Is the manuscript technically sound, and do the data support the conclusions?

Reviewer #1: Yes

Reviewer #2: Yes

2. Has the statistical analysis been performed appropriately and rigorously? 

Reviewer #1: Yes

Reviewer #2: Yes

3. Have the authors made all data underlying the findings in their manuscript fully available?

Reviewer #1: Yes

Reviewer #2: Yes

4. Is the manuscript presented in an intelligible fashion and written in standard English?

Reviewer #1: Yes

Reviewer #2: Yes

5. Review Comments to the Author

Reviewer #1: Authors conclude hypovascular tuomrs developed into HCC at a high rate despite the elimination of HCV by DAAs. There are no major revisions required. Minor revisions required are as follows:

#1: It is recommended to give a definition of how many millimeter or more the growth of hypovascular tumors should be.

#2: The pixels in Figures are a little bit coarse.

Reviewer #2: This is a manuscript entitled “Hypovascular tumors developed into hepatocellular carcinoma at a high rate despite the　elimination of hepatitis C virus by direct-acting antivirals

” by Kazuaki Tabu1, et al.

In this study, the authors analyzed whether DAAs can suppress the growth and hypervascularization of hypovascular tumors in prospective study. The authors showed that

of 33 hypovascular tumors, twenty of 33 nodules (60.6%) showed tumor growth or hypervascularization, and 13 (39.4%) nodules showed no change or improvement. The cumulative incidence rates of tumor growth or hypervascularization were 21.2% at 1 year, 36.8% at 2 years, and 55.9% at 3 years, and 69.9% at 4 years. And in patients with SVR, the cumulative HCC development rates of patients with hypovascular tumors was significantly higher than in

those without any tumors.

Although, this topic was analyzed by another groups, however, this study have several strength, including prospective study, and longer observation period.

This reviewer have some concerns to be address by the authors, as below

1. Please show whether the survey methods (CT/MRI vs echo) before treatment affect incidence of HCC or not,

, and please described when this survey was conducted (all patients were surveyed at initiation point of DAA?)

2 In Materials, and Method, Patients; the authors described that “ 168 without a hypovascular tumor underwent CE-MRI or CT within 2 year before therapy or during DAA therapy”

This reviewer think that 2 years is too long for diagnosis of no hypovascular tumor at baseline.

To avoid this, please limit the patients who were conducted CT around the point of initiation of DAAs, or to described this in limitation of this study.

3 In figure2, please show the analysis in patients who achieved SVR only.

6. PLOS authors have the option to publish the peer review history of their article (what does this mean?). If published, this will include your full peer review and any attached files.

Reviewer #1: Yes: Hidehiro Kamezaki

Reviewer #2: No

---

## [Author Response · Author response to Decision Letter 0]

10 Jul 2020

Thank you for your valuable feedback. We have revised the manuscript based on the comments. 

Reviewer #1 

1. It is recommended to give a definition of how many millimeter or more the growth of hypovascular tumors should be.

Thank you for your comments. HCC development was defined by 20% growth or hypervascularization of a hypovascular nodule. We have added this to the Materials and Methods section. 

2. The pixels in Figures are a little bit coarse.

We have improved the resolution of the figure. 

Reviewer #2: 

1. Please show whether the survey methods (CT/MRI vs echo) before treatment affect incidence of HCC or not, and please described when this survey was conducted (all patients were surveyed at initiation point of DAA?)

The patients who underwent CT or MRI before DAA therapy had higher HCC development rates than those who underwent US alone. Because the patients who underwent CT or MRI more frequently had liver cirrhosis and a history of HCC therapy. We have created S1 Table and described the following in the Results section.

The patients who underwent CT or MRI before DAA therapy had higher rates of HCC development than those who underwent US alone. Because the patients who underwent CT or MRI more frequently had liver cirrhosis and a history of HCC therapy, had higher total bilirubin, GGT, hyaluronic acid, Fib-4 index, and AFP values before therapy and at the end of treatment, and had lower platelet counts and albumin levels than those who underwent US alone (S1 Table).

This study was a prospective observational study conducted in 29 patients with hypovascular tumors that were confirmed by CE-MRI or CT. Informed consent was obtained after the confirmation. Some of the patients applied for a medical subsidy program in Japan after giving their consent. Therefore, most patients were surveyed before DAA therapy. These surveys were conducted approximately 50 days before therapy. 

We have described this in the Materials and Methods section. In addition, 452 patients received DAA therapy after they were confirmed to be tumor-free by US, CT, or MRI. However, we did not confirm the exact US survey points before therapy. Thus, the initiation of the observation period was defined as the initiation of DAA treatment. We have described this as a limitation of the present study. 

2. In Materials, and Method, Patients; the authors described that “ 168 without a hypovascular tumor underwent CE-MRI or CT within 2 year before therapy or during DAA therapy”

This reviewer think that 2 years is too long for diagnosis of no hypovascular tumor at baseline. To avoid this, please limit the patients who were conducted CT around the point of initiation of DAAs, or to described this in limitation of this study.

Thank you for your comments. I agree with the reviewer's suggestions. Unfortunately, there were few patients who underwent CT or MRI immediately before DAA therapy. Since it overlaps with the comments in 1, we have described the following as a limitation of this study. 

Fourth, 29 patients had hypovascular tumors that were confirmed by CE-MRI or CT. These surveys were conducted approximately 50 days before therapy. Further, 452 patients received DAA therapy after they were confirmed to be tumor-free by US, CT, or MRI. However, we did not confirm the exact US survey points before therapy. Thus, the start of the observation period was defined as the time at which DAA treatment was initiated. In addition, 168 patients without a hypovascular tumor underwent CE-MRI or CT within 2 years before therapy or during DAA therapy. Unfortunately, there were few patients who underwent CT or MRI immediately before DAA therapy. We cannot rule out the possibility of growth or hypervascularization of hypovascular tumors, or HCC development in patients who were tumor-free before DAA therapy. Thus, a future study should analyze a larger cohort which imaging surveillance was performed immediately before DAA therapy.

3. In figure2, please show the analysis in patients who achieved SVR only.

We have shown the analysis in patients who achieved an SVR only in Figure 2 and 3. We therefore corrected the data in the Abstract, Results and Discussion.

---

## [Decision Letter · Decision Letter 1]

28 Jul 2020

Hypovascular tumors developed into hepatocellular carcinoma at a high rate despite the elimination of hepatitis C virus by direct-acting antivirals.

PONE-D-20-16242R1

Dear Dr. Seiichi Mawatari:

We’re pleased to inform you that your manuscript has been judged scientifically suitable for publication and will be formally accepted for publication once it meets all outstanding technical requirements.

Kind regards,

Tatsuo Kanda, M.D., Ph.D.

Academic Editor

PLOS ONE

Additional Editor Comments (optional):

Reviewers' comments:

Reviewer's Responses to Questions

**Comments to the Author**

1. If the authors have adequately addressed your comments raised in a previous round of review and you feel that this manuscript is now acceptable for publication, you may indicate that here to bypass the “Comments to the Author” section, enter your conflict of interest statement in the “Confidential to Editor” section, and submit your "Accept" recommendation.

Reviewer #1: All comments have been addressed

Reviewer #2: All comments have been addressed

2. Is the manuscript technically sound, and do the data support the conclusions?

Reviewer #1: Yes

Reviewer #2: Yes

3. Has the statistical analysis been performed appropriately and rigorously? 

Reviewer #1: Yes

Reviewer #2: Yes

4. Have the authors made all data underlying the findings in their manuscript fully available?

Reviewer #1: Yes

Reviewer #2: Yes

5. Is the manuscript presented in an intelligible fashion and written in standard English?

Reviewer #1: Yes

Reviewer #2: Yes

6. Review Comments to the Author

Reviewer #1: (No Response)

Reviewer #2: In this revise paper, the authors replied to my comments properly.

This study is worth reporting .

Thank you for giving me a chance to review this valuable article.

7. PLOS authors have the option to publish the peer review history of their article (what does this mean?). If published, this will include your full peer review and any attached files.

Reviewer #1: **Yes: **Hidehiro Kamezaki

Reviewer #2: No

---

## [Editor Report · Acceptance letter]

3 Aug 2020

PONE-D-20-16242R1 

Hypovascular tumors developed into hepatocellular carcinoma at a high rate despite the elimination of hepatitis C virus by direct-acting antivirals. 

Dear Dr. Mawatari:

I'm pleased to inform you that your manuscript has been deemed suitable for publication in PLOS ONE. Congratulations! Your manuscript is now with our production department. 

Kind regards, 

on behalf of

Dr. Tatsuo Kanda 

Academic Editor

PLOS ONE